# A numerical evaluation of real-time workloads for ramp controller through optimization of multi-type feature combinations derived from eye tracker, respiratory, and fatigue patterns

Quan Shao[1]*, Kaiyue Jiang[1], Ruoheng Li[2]

1 College of Civil Aviation, Nanjing University of Aeronautics and Astronautics, Nanjing, Jiangsu Province, China, 2 School of Electronic and Information Engineering, Beihang University, Beijing, China

* shaoquan@nuaa.edu.cn

## Abstract

Ramp controllers are required to manage their workloads effectively while handling complex operational tasks, a crucial part of improving aviation safety. The ability to detect their instantaneous workload is vital for ensuring operational effectiveness and preventing hazardous incidents. This paper introduces a novel methodology aimed at enhancing the evaluation of the ramp controller's cumulative workload by incorporating and optimizing the feature combination from eye movement, respiratory, and fatigue characteristics. Specifically, a 90-minute simulated experiment related to ramp control tasks, using real data from Shanghai Hongqiao Airport, is conducted to collect multi-type data from 8 controllers. Following data construction and the extraction of multi-type, the workloads of all samples are categorized through unsupervised learning. Subsequently, supervised learning techniques are used to calculate feature weights and train classifiers after data alignment. The optimal feature combination is established by calculating feature weights, and the best classification accuracy is over 98%, achieved by the KNN classifier. Furthermore, numerical evaluation and threshold calculations for different workload levels are interpreted. It is promising to provide insights into future works towards human-centered data construction, processing, and interpretation to promote the progress of workload assessment within the aviation industry.

## 1 Introduction

In 2019, China achieved a significant milestone, i.e., transitioning control of apron operations at airports with annual passenger volumes exceeding ten million from air traffic control to dedicated apron control teams [1]. This shift of command responsibility for the movement of aircraft on the surface marks the emergence of apron control as a nascent force within China's transport airports. In the domain of airport operation and management, the ramp emerges as a pivotal sector, wherein ramp controllers are vested with the imperative to orchestrate air-ground operations, sequence departures, and foster a seamless collaboration with stakeholders,

**Data Availability Statement:** All relevant data are within the manuscript and its Supporting Information files.

**Funding:** This work was supported by the National Natural Science Foundation of China-Civil Aviation Administration of China Civil Aviation Joint Research Fund Project (U2233208). The funders were involved in supervision, idea formulation, manuscript concept development, manuscript review, and revisions.

**Competing interests:** The authors have declared that no competing interests exist.

including pilots, thereby ensuring an efficient utilization of ground resources. Throughout work shifts, ramp controllers have to maintain a high level of management towards their performance while performing various control tasks in a complex information interaction environment. However, there is currently a growing concern about the increased risk of airport incidents caused by ramp controllers experiencing a higher accumulated workload [2]. Therefore, a real-time monitoring system seems required to assist airport managers in efficiently understanding and further managing the ramp controllers, particularly during periods of increased workload.

## 1.1 Workload measurement

With the rapid development of neuro-ergonomic and bio-sensor techniques, comprehending physiological responses is currently the most widely used method for gaining insight into and assessing personal workload states. Various signals have been used as indicators of workload or input features for machine learning classifiers, including cardiovascular signals, brain activity, and eye movements [3,4]. Recent studies suggest that incorporating personnel's respiratory data is also beneficial for workload evaluation, as respiratory features have been positively correlated with workload [5]. For example, He et.al [6] explored the impact of cognitive workload on respiratory rate in vehicle drivers. Eye movements, which reflect fluctuations in workload related to visual task requirements, are particularly valuable in complex work situations such as ramp control. This has been demonstrated in research within the civil aviation domain [7], highlighting their effectiveness in evaluating ramp controller workload.

On one hand, the neuro-physiological metrics were used to assess workload. For example, the electrocardiogram (ECG) was widely employed to monitor cardiovascular signal, providing necessary information about the cardiac cycle [8–10]. Meanwhile, the electroencephalography (EEG), including power spectrum density and event-related potentials (ERP) information, is the primary method used to quantify brain activity for correlating workload states [11,12]. Their findings indicated that higher cognitive workload was associated with increased respiratory rates. It is worth noting that ECG and EEG necessitate affixing electrodes to the participant's body or brain, demanding substantial preparatory work prior to data collection, proving intrusive for individuals involved, rendering these methods suitable primarily for simulation experiments and theoretical research. The respiratory sensors used to measure the respiratory waveform are typically compact and minimally disruptive to common work, serving as a non-intrusive indicator, effectively capturing the level of stress. Therefore, the practical value of these methods (except for the respiratory techniques) is limited during real-world situations.

On the other hand, unlike measurements that necessitate physical attachment to the user, eye movement is executed by wearable or remote devices that utilize high-precision cameras to capture images of the user's eyes.

Specifically, pupil size has been extensively studied [3,4] as it is considered a useful indicator of mental effort or processing load. Several studies have demonstrated a positive correlation between pupil size and workload using different methods and in various scenarios [13–16]. Gaze-related parameters of eye movements have been recognized as useful indicators of workload assessment. However, there is no consensus on the trends of these indicators. For instance, gaze duration either increases [17,18] or decreases as workload increases [15,19–21]. Multiple studies have confirmed that parameters related to gaze, such as gaze point, gaze duration, and gaze fixation position, have a significant impact on workload [16,22]. Furthermore, eye movement metrics related to the area of interest [23] and blinking [22,24] have also demonstrated promising results in workload analysis. However, according to Niezgoda et al. [14],

gaze position, gaze duration, and blink frequency are less sensitive indicators when measuring changes in cognitive demands in auditory-verbal-vocal tasks.

## 1.2 Workload measurement considering fatigue

Fatigue is the feeling of exhaustion and tiredness, and short-term fatigue is usually due to over-activity and lack of rest [25]. Workload refers to the amount of work that the body is subjected to per unit of time, including the energy consumed by the work and the difficulty and complexity of the work [26–29]. Fatigue is not equivalent to a high workload, but rather a result of the gradual accumulation of workload. Studies have shown that personnel fatigue is affected by their workload [30].

Real-time monitoring technology is widely used to assess personnel fatigue levels in road vehicle driving to advance the development of automatic driving technology. Previous studies on this technology can be divided into two primary areas: face detection technology and fatigue classification models. Face detection techniques involve various methods such as facial expression detection using OpenCV and Dlib libraries [31], and image key point detection and feature extraction using classifiers like convolutional neural networks and Adaboost combined with localized binary patterns [32–34]. These techniques are typically applied to real-time image or video streams [35]. The fatigue classification models can be further subcategorized into two classification methods: a priori and non-priori. Prior fatigue classification models typically relate to the experiment protocol, using datasets to train classification networks. Their research primarily focuses on optimizing and improving the performance of classification models [32,36,37]. Non-prior classification models are not directly related to experiment design, using fuzzy inference [33], numerical weighting analysis [34], and other techniques to determine levels of personnel fatigue. The duration, percentage, and frequency of eye closures are commonly used facial features to measure employees' levels of fatigue in real-time [33,34,36,38]. All of the suggested methods have a detection accuracy of over 90% [31,37,39]. Real-time fatigue monitoring methods have been shown to be effective in promptly and accurately determining the level of driver fatigue during road driving tasks. **Table 1** provides a concise overview of the most significant and pertinent findings from the aforementioned literature.

The extant literature indicates that physiological indicators such as cardiovascular signals, brain activity, and eye movements are frequently employed for the assessment of personnel workload, with the general consensus being that they provide reliable indications. The aforementioned studies have demonstrated that these signals, in conjunction with respiratory data and real-time fatigue detection methods such as facial expressions and eye closures, can effectively gauge workload and fatigue levels. However, these methods have notable limitations. The validity of these methods is variable across different tasks and individuals, which presents a challenge to their universal application. Many current approaches rely heavily on subjective evaluations, which may not capture transient fluctuations or provide real-time insights critical for dynamic environments. Furthermore, existing techniques often offer only static snapshots of workload rather than continuous, real-time monitoring, which is essential for environments with rapidly changing conditions, such as ramp control. To address these issues, a more robust system must be developed that accommodates individual variability, provides continuous workload insights, and improves real-time monitoring capabilities to enhance practical application in high-stress scenarios.

## 1.3 Challenges, objectives, and contributions

Physiological features have been commonly used as a well-established measure to assess personnel workload in various fields and generally demonstrate high accuracy [40]. However,

**Table 1. Literature summary.**

| Study | Methodology and Performance | Focus | Key Findings |
|---|---|---|---|
| Charles et al. (2019) [3] | Analysis of physiological signals (cardiovascular signals, brain activity, and eye movements) with 85% predictive accuracy in workload assessment | Workload assessment through cardiovascular signals, brain activity, and eye movements | Identified various physiological signals as reliable workload predictors |
| Tao et al. (2019) [4] | Eye movement tracking in complex tasks, achieving 80% accuracy in distinguishing high vs. low workload conditions | Workload evaluation in complex tasks | Demonstrated effectiveness of eye movements in workload assessment |
| Liu et al. (2019) [5] | Respiratory data analysis, with a 0.75 correlation coefficient between respiratory features and workload | Correlation between respiratory features and workload | Positive correlation between respiratory features and workload |
| He et al. (2019) [6] | Analysis of cognitive workload's impact on respiratory rate, showing a 10–15% increase in respiratory rate with increased cognitive workload | Effect of cognitive workload on respiratory rate | Increased cognitive workload associated with higher respiratory rates |
| Wen-Chin Li (2012) [7] | Eye movement analysis for ramp controllers, with 90% accuracy in identifying high workload scenarios | Workload evaluation for ramp controllers | Eye movements are valuable for assessing ramp controller workload |
| Savas et al. (2020) [31] | Facial expression detection using OpenCV and Dlib for real-time fatigue monitoring, achieving 88% detection accuracy | Real-time fatigue detection | Effective facial expression detection for fatigue monitoring |
| Li et al. (2019) [32] | Image key point detection and feature extraction for real-time fatigue monitoring, achieving 87% classification accuracy | Real-time fatigue monitoring | Utilized key point detection for fatigue classification |
| Liu et al. (2020) [33] | Fuzzy inference for non-priori fatigue classification, with 82% accuracy without prior knowledge of fatigue patterns | Non-priori fatigue classification | Non-priori models for determining fatigue levels |
| Zhu et al. (2022) [34] | Numerical weighting analysis for non-priori fatigue classification, achieving 85% classification accuracy | Non-priori fatigue classification | Effective use of numerical weighting for fatigue classification |
| Cheng et al. (2019) [38] | Eye closure duration and frequency measurement, detecting fatigue with 92% accuracy based on eye closure frequency and duration | Real-time fatigue assessment | High accuracy in detecting fatigue through eye closures |

their validity may not be universal in all task scenarios [3,4]. Furthermore, even when individuals perform similar tasks in the same environment, their physiological data may vary significantly [41–43].

In addition, some physiological responses are used as input features, instead of serving as indicators, to construct predictive models of personnel workload status. Cardiovascular signals [44,45], brain activity [45–47] and eye movements [24,48] remain the three most common types of features. Algorithms commonly used for constructing model include support vector machines (SVM) [24,47,48], convolutional neural networks (CNN) [46,47], two-stream neural networks (TSNN) [46,47], Long Short-Term Memory Neural Networks (LSTM) [45], and Artificial Neural Networks (ANN) [44], among others, with classification prediction accuracies typically ranging from 70% to 90%. These studies typically rely on subjective evaluations, such as the NASA Task Load Index (NASA-TLX) or the n-back experimental paradigm, as labels for the data samples. It is worth noting that correlation analyses that do not use machine learning algorithms may also employ subjective methods. However, it has been found that subjective ratings may not capture more transient fluctuations in the system or in the level of workload during display interactions [8]. This limitation significantly hampers the application of these assessment methods in critical ramp control workload management scenarios, where the necessity for real-time performance cannot be overstated. In practical settings, forecasting personnel workload with precision presents a formidable challenge. The development of supervised classifiers necessitates the extensive collection of data on personnel work behavior and pertinent traffic details during the initial stages of system design. Moreover, these methods struggle to accommodate the variances in individual responses inherent among ramp controllers. Additionally, current workload assessment techniques primarily yield ratings of personnel workload, which, while informative, do not afford airport managers insights into the dynamic and evolving nature of workloads or the cumulative conditions affecting personnel.

Consequently, the practical implementation of such systems primarily offers a snapshot of current load ratings, failing to capture the nuanced and fluctuating aspects of ramp controllers' workloads.

In this study, the objectives are to (i) introduce real-time fatigue state as an additional primary indicator (besides the other physiological indicators) and (ii) use a non-priori method to calibrate workload levels of the ramp controllers.

- introducing the real-time monitoring of the ramp controller's fatigue state as a workload characterization indicator and developing a real-time fatigue monitoring system based on facial and eye recognition. The accuracy of workload assessment could be improved by identifying multi-type feature combinations that can be used to calibrate the workload of ramp controllers. *(Objective 1)*

- proposing a numerical evaluation of workloads to determine the threshold for workload levels. The workload categories are classified based on non-priori experimental data collected from continuous control work, in which the unsupervised and supervised methods are used to realize the clustering labelling and classification. The classification performance and feature importance are used as the criteria for selecting feature combinations to improve the performance of workload evaluation. *(Objective 2)*

The expected findings will support the improvement of decision-making in workload management and the efficiency of the ramp control operations, facilitating the transition from theoretical analysis to practical application. The main contributions are summarized as below:

1. Development and implementation of a real-time fatigue monitoring system. Unlike previous approaches that rely on singular physiological indicators, this system employs a multifaceted analysis incorporating facial recognition technology to assess fatigue states. By analyzing a combination of physiological behaviors and integrating multiple indicators through complex algorithms and models, this system offers a more nuanced and comprehensive understanding of the controller's fatigue state. The system's capability to identify and utilize multi-type feature combinations (both fatigue and physiological data) further refines workload calibration, offering a significant leap in the precision of fatigue state monitoring and its implications for workload characterization.

2. Innovative non-priori method for hierarchical workload calibration. The research introduces a groundbreaking non-priori numerical transformation method to calibrate workload levels for ramp controllers. This method stands out by enabling continuous observation and nuanced classification of workload intensity without relying on predefined categories or thresholds. Utilizing unsupervised learning techniques to classify personnel load class at each time unit during continuous control tasks, the study transforms traditional unsupervised methods into a supervised framework under machine learning algorithms. This approach not only facilitates a dynamic and continuous assessment of workload but also enhances the recognition of potential safety risks by determining the threshold for each level based on numerical values. The method's efficacy is further underscored by its application in selecting feature combinations, guided by the results of classifiers and classification accuracy, thereby offering a novel and effective tool for workload calibration.

3. Advanced numerical assessment of workload. By integrating transient features such as fatigue, eye movement, and respiratory characteristics, collected from simulated ramp control tasks, into the workload assessment model, the study achieves an unprecedented classification accuracy of over 98%. This high level of precision is attained through meticulous

data sample analysis over continuous 90-minute periods, utilizing unsupervised learning for workload level determination and supervised techniques for feature weight computation and classifier training. The methodological innovation extends to the optimal selection of feature combinations, taking into account feature weights, classifier accuracy, ensemble construction techniques, and workload levels labeling. The establishment of numerical workload expressions and the calculation of thresholds for workload levels not only demonstrate the model's efficacy but also provide a valuable framework for future research aimed at refining workload assessment accuracy.

## 2 Methodology

Compared to other physiological features, such as EEG and ECG, the features from the eye tracker and respiratory provide reliable conclusions, have high measurement stability, and cause minimal disturbance to individuals' normal work [4]. Meanwhile, the physiological data may vary for different individuals. Consequently, this paper utilized eye tracker, respiration, and fatigue metrics (obtained from real-time fatigue monitoring system based on the facial and eye recognition) and described empirical experiments. The collected data was used to determine the workload levels and to calculate the weights of the indicators, which were used to determine the optimal feature combination and to propose a numerical representation of the different workloads.

### 2.1 Participants

In this experiment, eight ramp controllers, including one female, were recruited. These participants possessed one to three years of work experience and had an average age of 26 years. All participants had successfully passed the Civil Aviation of China Air Traffic Controller License Examination and held valid licenses. Notably, two of participants, one of whom was female, had additionally completed two years of systematic and professional training in control studies. They all have normal or corrected-to-normal vision, no hearing impairments, and were right-handed. The recruitment period for this study begins on April 21, 2023, and ends on April 26, 2023. Participants provided and signed written informed consent. Each participant received 200 RMB as a compensation for experimental time and efforts. Moreover, this study was also approved by the Institutional Review Board of the Faculty of Psychology at the Nanjing University of Aeronautics and Astronautics (NUAA). The individual in this manuscript has given written informed consent (as outlined in PLOS consent form) to publish these case details.

### 2.2 Experiment simulation

**2.2.1 Experiment design.** The experiment employed the sim4D ramp simulation control system [49] to replicate real-world operational situations for data collection. In this simulated situation, based on the Hongqiao Airport, the controller interacted through voice commands with an integrated AI captain to execute all control operations, lasting around 90 minutes. The interface of the simulated control system is depicted in **Fig 1**. The experimental setup featured two monitors with a resolution of 1920x1080: a larger screen positioned directly in front of the participant, displaying the simulation control system, and a smaller screen situated to the participant's left, displaying the real-time fatigue monitoring system (*see Section 2.3.1*). The dimensions and distance of the monitors utilized by the ramp controllers in this experiment are consistent with those observed in the actual work environment.

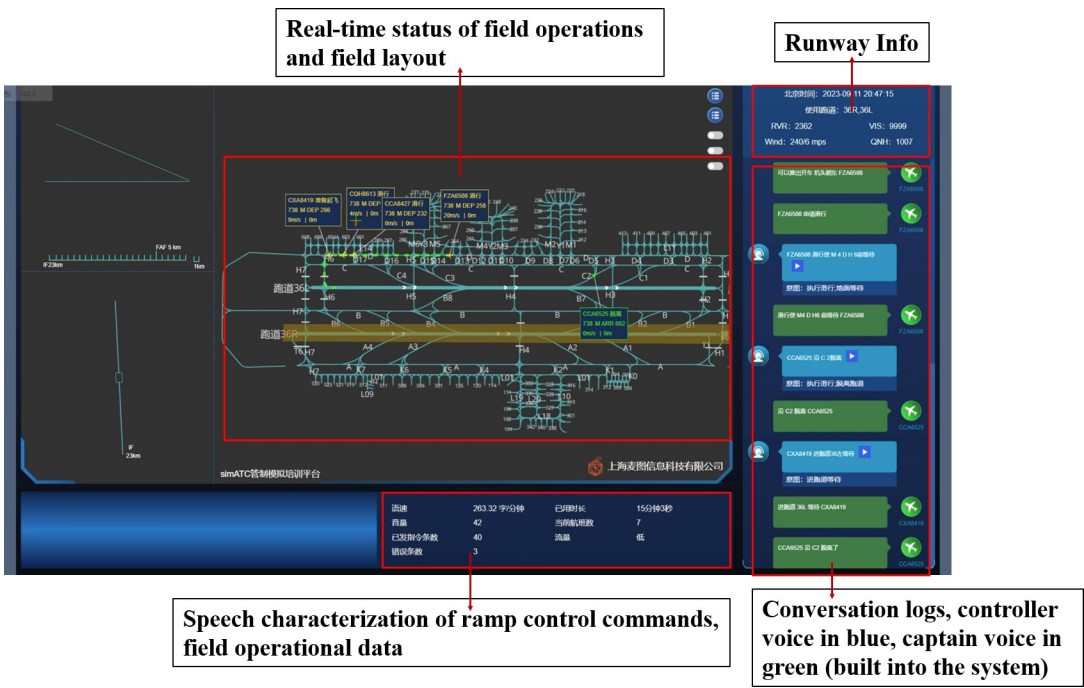

**Fig 1. The interface of the simulated control system, showing the single-runway operation of the west runway of Shanghai Hongqiao Airport and the simulated control and command scene on the ramp.**

**2.2.2 Apparatus.** A Logitech C922PRO HD camera, strategically placed directly in front of the participant, was utilized to capture facial inputs for the real-time fatigue monitoring system. This placement was chosen to ensure optimal stability in eye recognition while avoiding interference with the visibility of the display screen. The experimental setup incorporated four laptops, each serving a distinct purpose: one dedicated to recording data from the physiological monitor, another for capturing eye movement data, a third tasked with operating the simulation control system and screen recording, and a fourth designated for managing the real-time fatigue monitoring system along with screen recording. Additionally, two cameras were set up to document the experimental proceedings in real time, as shown in **Fig 2**. The research team included an experimental instructor responsible for guiding the participants through the start and end of the experiment, operating the eye tracker's data recording software, and overseeing the real-time fatigue monitoring system. An assistant was also present to handle the physiological monitor's data recording and camera equipment.

Throughout the experiment, physiological data collection for ramp controllers was completed using the EYESO glasses, a lightweight head-mounted eye tracker weighing merely 47 grams and has a sampling rate of 380Hz. This device enabled the accurate tracking of pupil movements via the myopic ellipse, ensuring precise data acquisition. Additionally, the EYESO MultiPhy7, a multidirectional physiological recording monitor, was employed to capture the subjects' respiratory waveforms in real-time. The convenience of this device is notable; subjects were simply required to wear a sensor affixed to an elastic band around their waist. The setup of the experimental apparatus is depicted in **Fig 3**.

**2.2.3 Experiment procedure.** The experiment was conducted in the laboratory of Emergency Science and Technology at the College of Civil Aviation, NUAA. The space was well-lit and ventilated, and free from noise disruption. The consent forms from the participants were received prior to the experiment. The subject firstly sat on the simulated control seat, wore

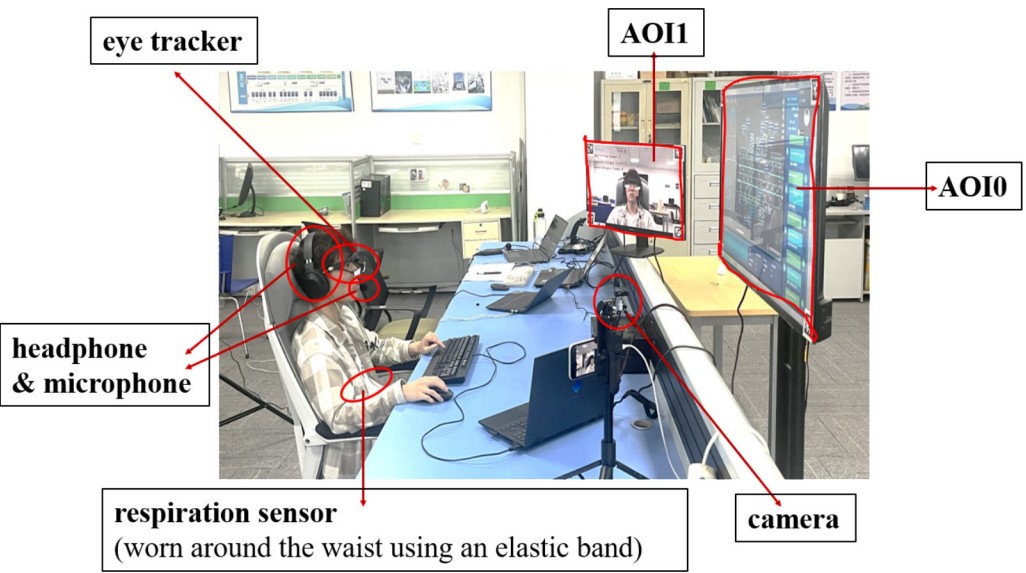

**Fig 2. The experimental setup, components, and area of interests (AOIs, see *Section 2.3*).**

physiological monitoring sensors, and kept their head still while wearing the eye tracker. The eye tracker was calibrated by the experimental instructor, who adjusted the settings of the pupil-recognition camera to achieve a confidence level of almost 100%. The assistant then prepared the camera equipment and analog control system for the pre-task start interface. Once

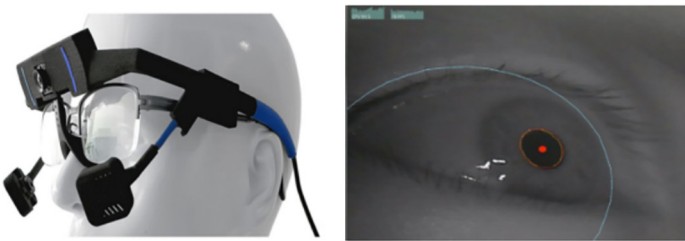

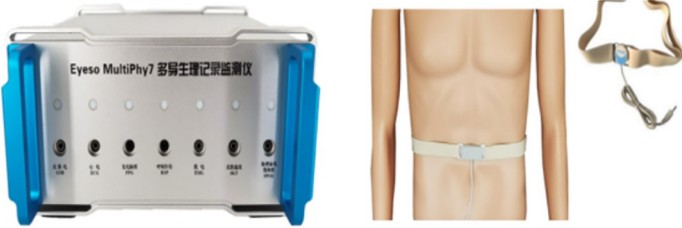

**Fig 3. Eye tracker and respiratory equipment.**

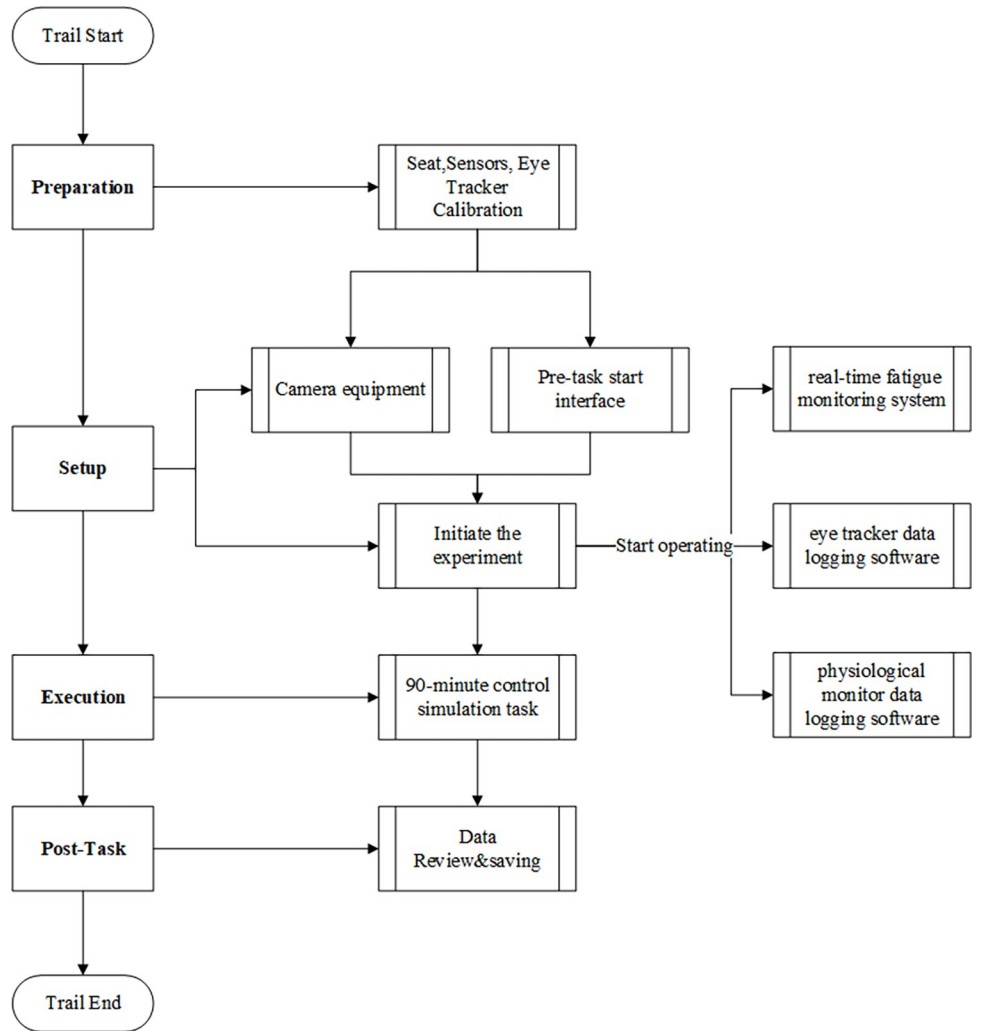

**Fig 4. Experiment flowchart.**

the experimental instructor initiated the experiment and concurrently operated the real-time fatigue monitoring system and eye tracker data logging software, the assistant simultaneously operated the physiological monitor data logging software. The participants carried out the 90-minute control simulation task autonomously without any external influence or disruption. The control task accurately reflected the actual flow and operational conditions of Hongqiao Airport. It involved a total of 71 flights for takeoff and landing. However, the specific changes in flight flow during the task were not known. Once the trial was over, each subject's data was reviewed and saved. The complete experimental procedure, including preparation, implementation, data collection, and review steps, is illustrated in **Fig 4**.

## 2.3 Measurements and indicators

**2.3.1 Real-time fatigue monitoring construction.** In terms of the real-time fatigue monitoring system, the facial and eye recognition were completed to assess fatigue levels in real-time. The method employs a camera to capture a real-time facial image. It then identifies the eyes and delineates the eye contour with a continuous green line displayed within a specified

window. The system's computational principle is based on the Percentage of Eye Closure (PERCLOS), which is calculated by determining the fraction of frames N with Eye Aspect Ratio (EAR) less than 0.3 in every 300 smoothed window frames, using the EAR as the base data. The system calculated the EAR and PERCLOS values for each frame and combined them using a weighting scheme to derive a comprehensive measure of ocular characteristics, denoted as M. These metrics were used to accurately represent workload. The equations used to calculate PERCLOS (Eq (1) and M (Eq (2)) is as follows:

$$PERCLOS = \frac{N_{EAR<0.3, in\ 300\ frames}}{300} \qquad (1)$$

$$M = 0.3*EAR + 0.7*PERCLOS \qquad (2)$$

The fatigue level is determined based on the number of successive frames with M greater than 0.605. The fatigue calculation approach is derived from the research of Zhu et al. [34]. The number of times M > 0.605 (i.e., the cumulative length of the driver's eye closure within a certain period) was recorded as consecutive frames (denoted by N). The following definitions were used to categorize the level of fatigue: $5 < N <= 15$ indicates mild fatigue, $15 < N < 50$ indicates moderate fatigue, and $N >= 50$ indicates severe fatigue. The number of occurrences for each fatigue level is recorded and displayed in real-time on the right side of the window. To ensure safety during operations and commands, we have added an alarm function to the system that alerts controllers to resume their duties promptly. The function is automatically activated when the number of consecutive frames reaches or exceeds 50, indicating a SEVERE level of fatigue, with a red alarm message appearing at the top of the interface. The system will record the frequency and length of each alarm message, and all the recorded data will be stored in a timestamped CSV file generated by the system. The facial fatigue recognition software used in this study was custom-developed by the research team. **Fig 5** shows the system interface.

The eye recognition system in the research is adversely affected by the eye tracker's simultaneous requirement to capture eye movement features. The dark portion of the eye tracker

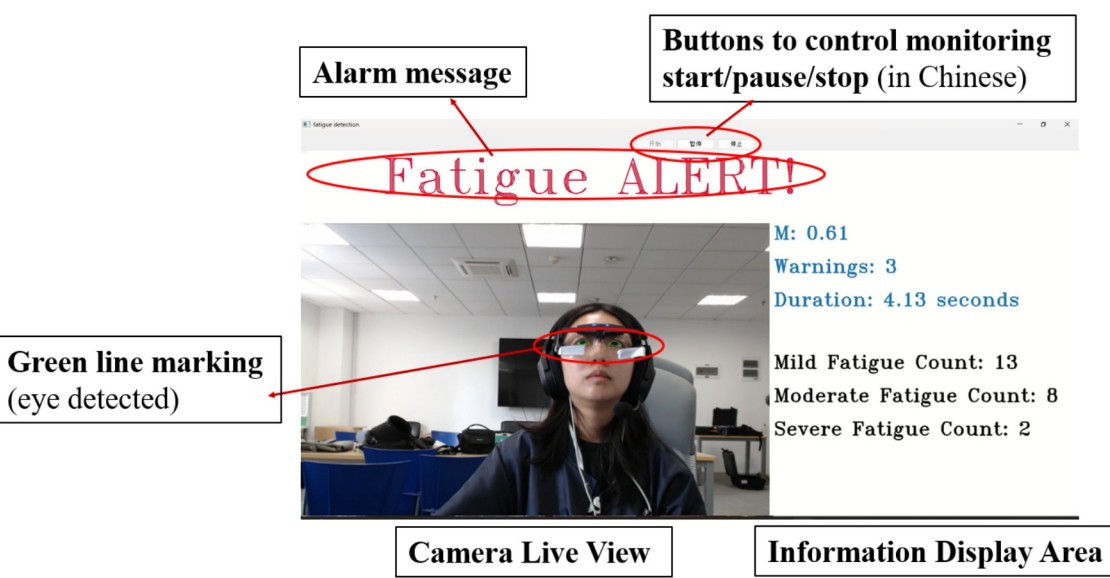

**Fig 5. Real-time fatigue monitoring system interface.** The number of fatigue level occurrences is updated after the alarm message disappears.

usually interferes with the algorithm's assessment of the pupil within the video frame. In addition, wearing an eye tracker prevents light from reaching the eye area, creating a dark region that hinders the system's ability to detect. Consequently, the recognition rate is notably lower than when the eye tracker is not worn. However, the system exhibited an eye identification rate of over 99% during the pre-experiment system test. This allowed it to instantly capture the appearance of the eye region of the face within the camera's shooting range. During the experiment, we covered the front black region of the eye tracker that was visible to the camera with white paper. We also positioned light sources around the subjects to enhance the illumination and brightness of their facial images. To clarify, although the eye tracker had a detrimental effect on the system's recognition rate, the number of missing frames remained within an acceptable range, and the detection stability remained high.

To address missing frames, we processed the data into unit time samples using the following approach: for the metrics EAR, PERCLOS, and M, we averaged the remaining samples after removing missing values if the proportion of missing frames was below 50% in a given unit of time. If the proportion exceeded 50% in that unit of time, we used random forest regression [50] interpolation to interpolate the results. The interpolation results were manually reviewed, corrected, and then averaged across all the data for that unit of time. The missing data for occurrences of fatigue levels were not considered, and the highest value within a specific time period was selected.

**2.3.2 Physiological indicators.**   Physiological indicators were also used to recognize workloads, and we utilized eye movement and respiration features for the fusion assessment besides the above fatigue indicators. The eye movement metrics were chosen based on previous studies that used features related to pupil change, blinking, gaze, sweep, area of interest gaze, and shifting [16,21,22,51–53]. **Table 2** displays all 20 indicators that were selected before the data collection phase of this study.

**Table 2. Indicators used in the data collection phase.**

| NO. | Feature Category | Specific Indicators (unit time: 1 minute) |
|---|---|---|
| 1 | real-time fatigue | EAR average |
| 2 | | PERCLOS average |
| 3 | | M average |
| 4 | | Mild Fatigue occurrences |
| 5 | | Moderate Fatigue occurrences |
| 6 | | Severe Fatigue occurrences |
| 7 | eye movement | average pupil diameter |
| 8 | | count of blinks |
| 9 | | average gaze duration |
| 10 | | average sweep length |
| 11 | | total number of gaze points |
| 12 | | total number of sweeps |
| 13 | | average gaze duration |
| 14 | | average sweep duration |
| 15 | | average gaze duration in AOI |
| 16 | | percentage of time spent gazing in AOIs |
| 17 | | number of gaze transfers between AOIs |
| 18 | | number of gaze transfers between AOI and non-AOI |
| 19 | respiration | respiratory rate |
| 20 | | average respiratory amplitude |

We also labeled the portion of the display showing the ramp simulation control system as Area of Interest 0 (AOI0), and the screen showing the real-time fatigue monitoring system as Area of Interest 1 (AOI1).

## 2.4 Data analysis

After data collection from the above experimental simulation, we divided the samples into 1-minute segments, the dataset consists of 720 data samples (in *dataset*) from 8 subjects in total, containing all the metrics mentioned in **Table 2**.

**2.4.1 Data clustering–unsupervised learning.** A preliminary feature selection was performed after data cleansing, where metrics with minimal variation across the control task were identified and eliminated. The metrics for fatigue condition were fully preserved when converting the data from each frame into samples per unit of time.

In the preparation of the dataset for clustering analysis, three principal features were selected based on their relevance to the study objectives: fatigue, eye movement patterns, and respiration. Numerical data underwent normalization, entailing subtraction of the mean and division by the standard deviation, to ensure comparability across features. Categorical data, specifically string attributes, were transformed using one-hot encoding to facilitate their incorporation into quantitative analyses.

Subsequent to these preprocessing steps, Principal Component Analysis (PCA) was applied to reduce the dataset dimensionality to two principal components. This reduction was instrumental in preserving the intrinsic data variance essential for effective cluster delineation, while simultaneously enabling visual representation of the data space [54]. The combined use of normalization, one-hot encoding, and PCA thus constituted a comprehensive methodological framework for data preparation, tailored to enhance the clustering algorithm's efficacy through the focused analysis of fatigue, eye movement, and respiration features.

Finally, we determined the optimal number of categorical clusters (K) using Silhouette analysis [55]. The Silhouette score is a quantitative measure used to evaluate the effectiveness of clustering. It is determined by assessing the similarity of data points within their respective clusters compared to the nearest surrounding clusters. A higher Silhouette score, closer to 1, indicates better clustering performance. The data were classified using the K-means clustering algorithm. The workload level for each cluster was determined based on the data characteristics, assuming that workloads will generally increase progressively over time.

**2.4.2 Screening feature combination–supervised learning.** 20% of the data was designated as a distinct test set for the purpose of evaluating the accuracy of the predicted outcomes, while the remaining 80% was utilized for the training and validation processes. The Random Forest algorithm is used to calculate the importance value of the features for secondary feature selection, with an initial five-fold cross-validation employed to identify the optimal features, which were then subjected to a second five-fold cross-validation. This dual approach optimizes model performance while conserving computational resources. The separate test set provides an independent measure of the model's generalization capability, ensuring robust and unbiased performance evaluation. Clustering classification is implemented prior to this to enhance the comprehensive involvement of personnel behaviors in the classification process and to improve the scientific validity of workload level labeling. This feature selection approach involves using workload level labels to train the Random Forest algorithm and compute the significance values of the features. Prior to this step, it is necessary to map workload levels to the samples to meet the requirements of the supervised algorithm input. The method also aims to determine the weight values of the features to compute the final load value and perform threshold analysis.

Only features with significance values of 0.010 or higher were retained. In addition, a separate test set comprising 20% of the dataset was randomly selected for model comparison. The remaining data was used for 5-fold cross-validation. The study utilized six basic algorithms: support vector machine (SVM) [56], decision tree (DT) [57], random forest (RF) [50], K-Nearest Neighbors Algorithm (KNN) [58], logistic regression (LR), and neural network (NN). The models have shown robust performance in managing small data samples, are easily understandable and interpretable, and are effective in addressing nonlinear relationships and multi-category problems [59]. Furthermore, four integrated learning techniques, namely voting [60], stacking [61], bagging [62] and boosting [63], were applied to the six base models mentioned above to create four integrated models. The hyperparameters for each model are determined through a grid search. This step converts the learning process from unsupervised to supervised, evaluates the performance of multiple classifiers, and selects the most appropriate learning method. A basic model for workload classification evaluation is constructed, and the accuracy of the classification results is used to assess the effectiveness of different feature combinations in characterizing the workload. Additionally, to enhance result reliability, 10 iterations were conducted for each feature set, and the classification accuracy results with the highest frequency were selected. This methodology balances model optimization with rigorous evaluation, even with a relatively small dataset.

To obtain a lower dimensional feature sample set, increase the threshold for feature selection to 0.040. Additionally, use the Featuretools toolkit [64] to determine the optimal feature engineering method and generate cumulative features, which have better performance in tasks that reflect time series and cumulative effects. By inputting these two mentioned feature sets into the selected classifier algorithm from the previous phase, we compare the classification performance to determine the most effective workload representation.

**2.4.3 Numerical workload evaluation.** After selecting the optimal combination of workload characterization features, each feature in each time unit is transformed into a numerical result using feature importance weights. The calculation formula is: (Eq (3))

$$Workload = \sum_{i=0}^{n} c_i V_i \tag{3}$$

Where denotes the value of each feature in the optimal set (the value after transformation according to the feature set construction method), is the feature weight (if the optimal feature set is a cumulative sum of features, the cumulative sum of features uses the same weight as the temporal features), and i = 1, 2,. . .n is the number of features included in the optimal set.

The threshold for each workload grade was determined by observing the cumulative changes in workload values over time and associating them with the results of the grading assessment. Because the maximum workload of ramp controllers during a work shift is often unknown to airport managers, this study relies on the absolute values obtained from the calculations to determine the workload grade thresholds, rather than performing additional processing, such as converting the workload values to a percentage. The enumeration method is used to determine sample values that have the same time sequence and are sorted in the same percentage position in the forward and reverse order in the adjacent groups after the samples in the group are sorted by time, respectively, as the threshold for dividing the workload level of that adjacent group within the time overlap of different workload levels.

Python 3.9 [65] and Jupyter Notebook [66] are used to execute the data processing and machine learning procedures. **Fig 6** depicts the sequential actions and procedures involved in the primary techniques used in the data analysis phase.

# 3. Results & discussion

## 3.1 Real-time fatigue monitoring system

This study used individuals' real-time fatigue status as an index for workload characterization. An independent real-time fatigue monitoring system was constructed to collect these data. Due to experimental constraints, the system exhibited an average detection omission rate of approximately 6%. However, since the indicators reflecting the real-time fatigue condition are continuous and stable, the collected data still fall within an acceptable range for handling missing values and errors.

The short-term system monitoring was initially observed. **Fig 7** illustrates the variations in EAR, PERCLOS, and M from frame 5001 to frame 5018, which corresponds to approximately the $32^{nd}$ minute of the experiment. According to the data trend, an EAR value below 0.25 suggests personnel blinking and can be detected by the system. PERCLOS is a value calculated using a smoothed-window technique, resulting in a more gradual trend. The alteration in M values is influenced by the collective impact of the EAR and PERCLOS patterns. These statements are consistent with the computational principles incorporated into the fatigue monitoring system developed in this study and suggest that the system operates as intended and can be implemented.

The fatigue status of the personnel during the control task was monitored in real-time. The statistics showed that all participants experienced varying degrees of fatigue within the initial 10 minutes of work. The subjects' EAR, PERCLOS and M values were averaged under the fatigue monitoring system with an alarm function to obtain **Fig 8.** The three measurements showed a consistent pattern, specifically increased levels of fatigue in the subjects at approximately 12 minutes into the task and when the task was almost half completed. There were consecutive instances during minutes 36–39 and 43–49 where the average value of M exceeded the critical threshold, indicating that the personnel may have been experiencing an overload condition. However, in contrast to expectations, during the second half of the task, there were no instances where the average value of M exceeded the critical value, indicating a decrease in personnel fatigue. This could be attributed to fluctuations in flight traffic and the volume of incoming and outgoing flights during the control task. After experiencing increased fatigue, individuals may become less patient and more easily distracted, which can contribute to the alleviation of the fatigue state.

In the present study, we conducted a comparative analysis of two sets of features: those that indicate fatigue (such as EAR and PERCLOS) and those that do not. The incorporation of these fatigue-related variables facilitated the discernment of operator workload, particularly in the identification of potential overload conditions. While these indicators may not encompass underload scenarios, such as drowsiness, they offer substantial insight into the correlation between fatigue and workload in high-demand circumstances. Subsequent research will investigate the influence of drowsiness and its potential implications for underload conditions, thereby extending the scope of these findings.

## 3.2 Feature selection

This study utilized three main factors to characterize workload: real-time fatigue status, personnel eye movement characteristics, and respiration features. Two feature selections were made during the data processing stage.

**3.2.1 Preliminary screening.** In the preliminary screening phase, we evaluated eye movement metrics and respiratory features based on their variance during the control task. As a result of the minimal variance observed in certain metrics, such as pupil diameter and the number of

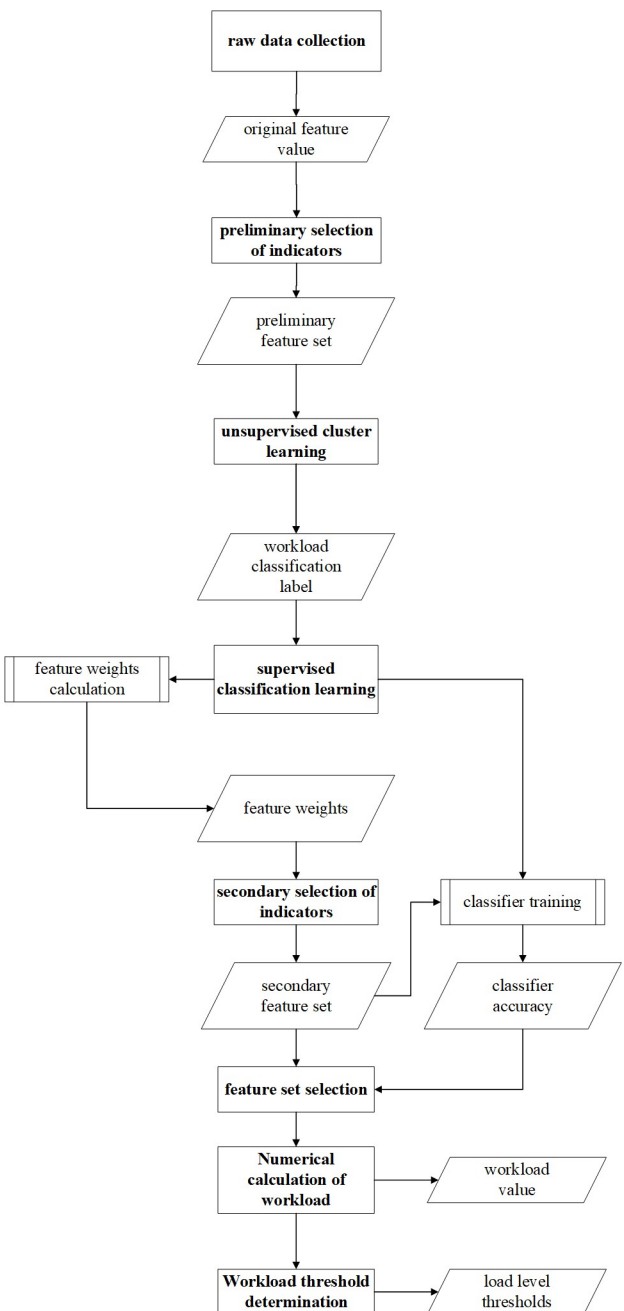

**Fig 6. Data analysis procedures.** Workload is recognized by eye tracker, respiratory and fatigue, and the raw data collection section contains data from all three categories.

gaze points, when the time axis was compressed, these were excluded from further analysis. Conversely, metrics related to gaze areas, including blink count and the percentage of time spent gazing in AOIs, were retained due to their significant variation and importance for analyzing eye movement features. Regarding respiratory characteristics, as respiratory amplitude remained highly stable, we elected to retain only the respiratory rate for subsequent analysis.

The oculomotor metrics were measured during the control task. The average pupil diameter ranged from 1.5 mm to 3.5 mm. The number of gaze points varied between 92 and 115,

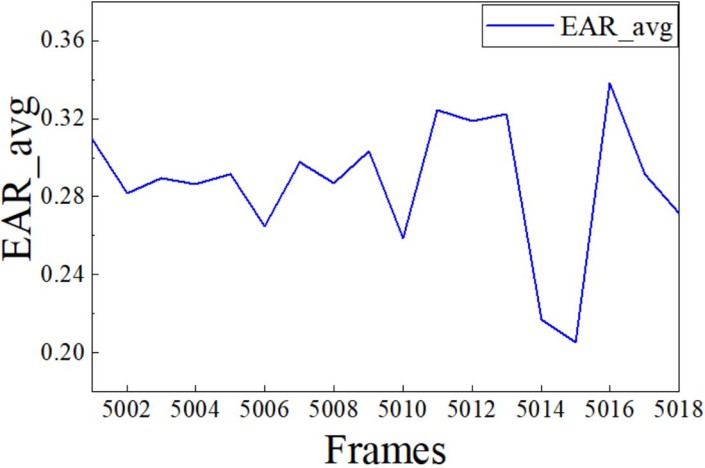

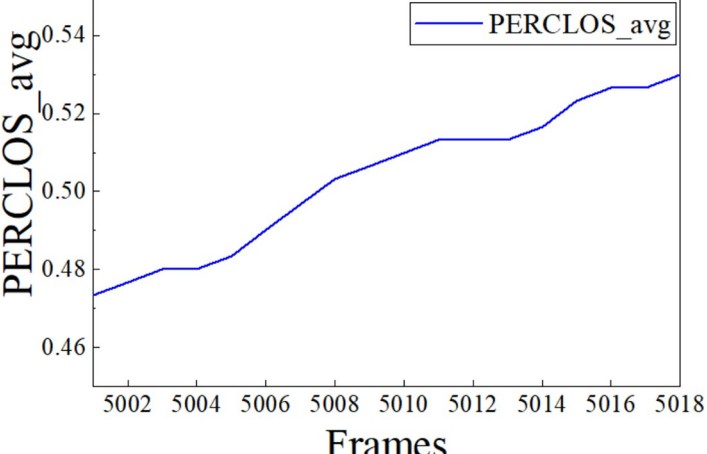

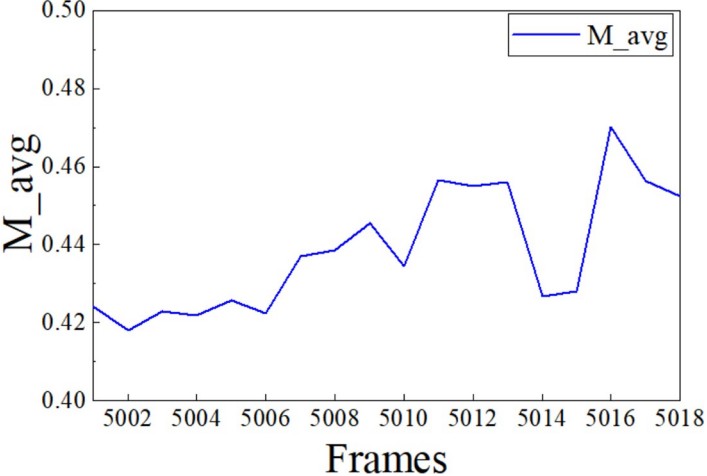

**Fig 7. EAR, PERCLOS, and M short-term variations.**

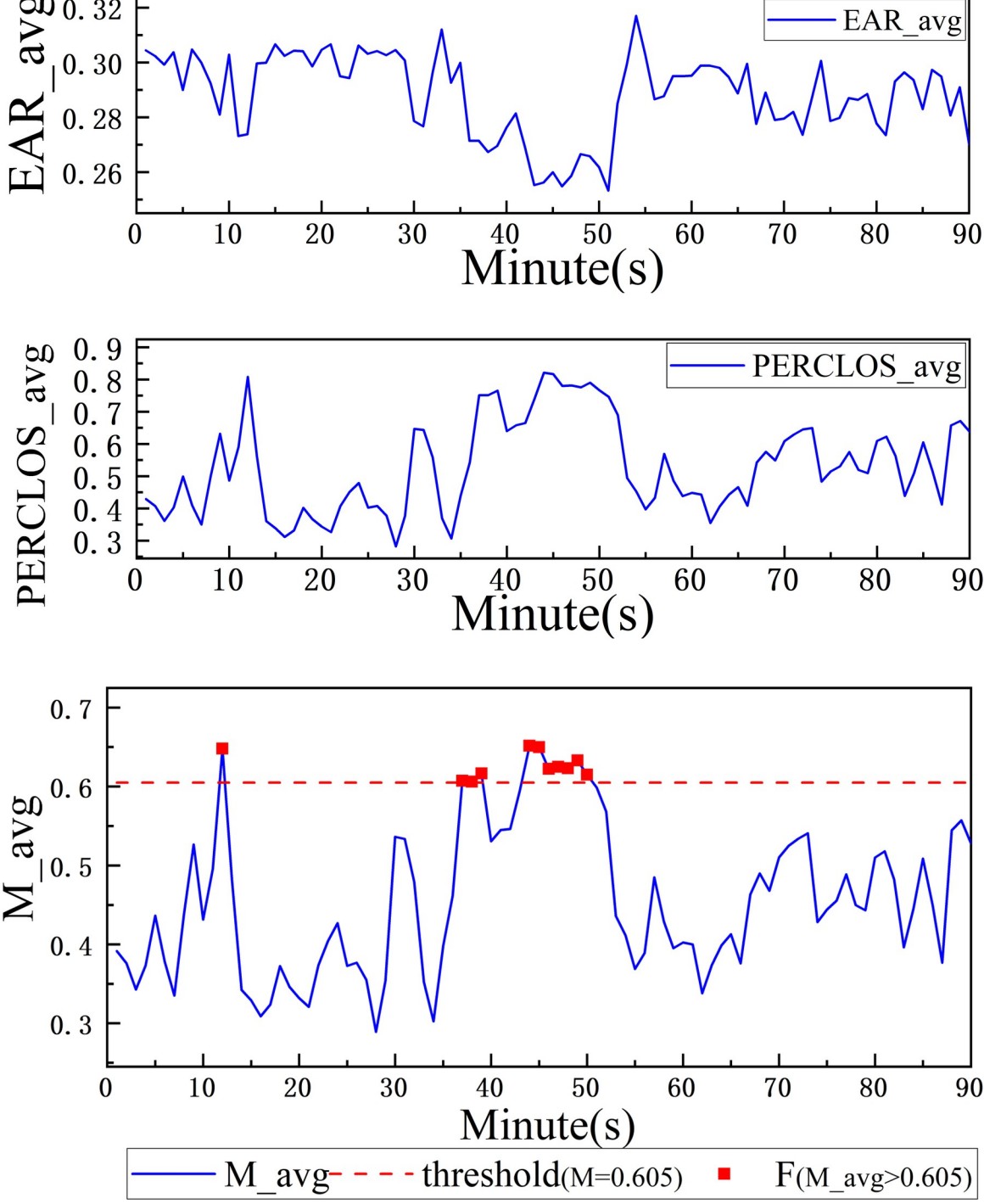

**Fig 8. EAR, PERCLOS, and M long-term variations.**

while the total number of sweeps ranged from 91 to 112. The average length of a single sweep varied between 131 and 185 pixels, and the average duration of a single sweep ranged from 76 to 90 milliseconds. The magnitude of variance in the values of these eye-movement measures was minimal and failed to demonstrate significant alterations upon compression of the time

axis. On the other hand, the average gaze duration ranged from 45,000 to 65,000 milliseconds, and the average gaze duration at AOI0 ranged from 30,000 to 65,000 milliseconds. Despite the considerable variability in these two measurements, we were unable to identify a discernible pattern of change. Additionally, the area-of-interest gaze ratio exhibited similar personnel behavior as these two metrics, but with more succinct data. Hence, we preserved the count of blinks, the percentage of time spent gazing in AOIs (AOI0 and AOI1), number of gaze transfers between AOIs, and number of gaze transfers between AOI and non-AOI, amounting to a total of five metrics, to analyze the eye movement features of the personnel.

Regarding respiratory characteristics, the study found that respiratory amplitude remained extremely stable throughout the control work and did not meet the study's requirements. The respiratory rate (number of respirations per minute) was retained for the next stage.

**3.2.2 Second screening.** For the second step of feature screening, we selected eight metrics with feature importance values greater than or equal to 0.010. These metrics are time (0.599), mild fatigue occurrences (0.203), number of gaze transfers between AOI and non-AOI (0.049), EAR average (0.047), moderate fatigue occurrences (0.046), percentage of time spent gazing in AOI0 (0.016), M average (0.014), and PERCLOS average (0.011). The objective was to maintain the sample characteristics while excluding any factors that could cause confusion. The feature importance values for different subjects and the activation status of the alarm function are all less than or equal to 0.001. This indicates that both different subjects and alarm systems have minimal impact on the categorization and assessment of the workload levels. This finding supports the validity of the experimental setup presented in this paper and the practicality of the real-time fatigue monitoring system. **Table 3** displays the final indicators for set construction.

## 3.3 Workload classification performance

**3.3.1 Unsupervised clustering results.** In this study, the feature selection process was structured into three main categories: real-time fatigue status, eye movement characteristics, and respiratory features. Each category was rigorously assessed to determine the most relevant metrics. Initial screening involved evaluating the variance of these metrics to identify those with significant and reliable contributions.

Subsequently, Principal Component Analysis (PCA) was applied to reduce the dimensionality of the selected features. PCA facilitated the transformation of the original feature set into

**Table 3. Indicators used in the feature combination phase.**

| Rank. | Specific Indicators (unit time: 1 minute) | Feature Importance |
|---|---|---|
| 1 | time | 0.599 |
| 2 | Mild Fatigue occurrences | 0.203 |
| 3 | number of gaze transfers between AOI and non-AOI | 0.049 |
| 4 | EAR average | 0.047 |
| 5 | Moderate Fatigue occurrences | 0.046 |
| 6 | percentage of time spent gazing in AOI0 | 0.016 |
| 7 | M average | 0.014 |
| 8 | PERCLOS average | 0.011 |
| 9 | count of blinks | 0.006 |
| 10 | respiratory rate | 0.002 |
| 11 | Severe Fatigue occurrences | 0.002 |
| 12 | number of gaze transfers between AOIs | 0.002 |
| 13 | percentage of time spent gazing in AOI1 | 0.001 |

two principal components, which encapsulated the primary variations in the dataset while reducing redundancy. These principal components were then utilized in the unsupervised learning phase, specifically through the K-means clustering algorithm. The K-means algorithm employed these two principal components to perform clustering.

Prior to selecting secondary features, we conducted a Silhouette analysis on the initial screened dataset. The results show that for k = 2, the score is above 0.57, while for k = 3–6, it remains consistently below 0.5. The study suggests that clustering is more effective when the number of clusters is limited to 2. Since a higher number of clusters is not useful for the practical applications of this study, we used the K-means clustering algorithm to bicategories the dataset, **Fig 9** illustrates the clustering phenomenon. This binary clustering approach effectively captured the workload levels and ensured that the results were both interpretable and applicable to practical scenarios.

The data samples were categorized based on clustering outcomes. The classification findings indicate that most samples located at the beginning of the experiment were classified as cluster 1, while those located closer to the end were categorized as cluster 0. To account for workload time-accumulation, we assigned a lower workload label to samples in cluster 1 and a higher workload label to cluster 0. However, the two clusters contained samples that were not completely separated in terms of time horizon. In fact, a few samples from the early stages of the experiment were categorized as belonging to the higher load condition, and vice versa. Unlike previous studies that predetermined the workload levels of the control task during the experimental stage or relied on subjective scales for mapping and direct evaluation and prediction using supervised algorithms, the use of unsupervised algorithms in this step offers several

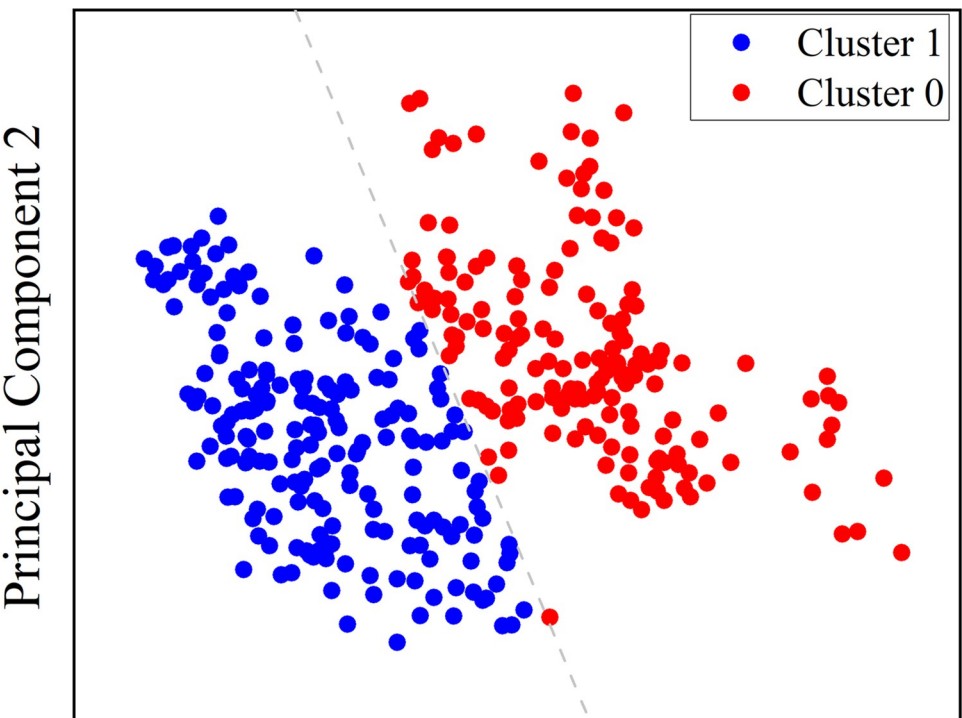

**Fig 9. Clustering result.** Cluster 0 represents high level and Cluster 1 represents low level of workload.

advantages. Firstly, it avoids the issue of unstable model accuracy caused by subjective load evaluation. Secondly, this aligns with real-world work scenarios where it is impossible to determine the load condition beforehand. The training difficulty of supervised machine learning classifiers will be significantly increased when taking into account the distinct individual variations across ramp controllers.

**3.3.2 Supervised classification results.** After performing clustering, the feature weights are computed, and a secondary feature selection is conducted. Then, the samples are fed into the classifier, using only the eight-dimensional features that were retained. Multiple classification tasks are carried out to mitigate the unstable accuracy issue caused by the sample data from the independent test set, which lacks strong generalization. The study collected a small dataset of 1676 samples, excluding any invalid data. A low complexity model was used due to the simplicity of the categorization task. The test results show that KNN and Bagging have superior performance with an accuracy of 98%. This is likely due to their enhanced adaptability to the data. The voting and stacking methods exhibit a high level of accuracy, approximately 97%, indicating the effectiveness of the integrated learning strategy. The Random Forest technique is quite effective in mitigating overfitting. The performance of boosting on the test set is quite low, with a classification accuracy of 93.1%. This could be attributed to potential overfitting. **Table 4** presents the accuracy of all learning models on both the cross-validation set and the test set.

Two models, RF and KNN, were selected based on their strong performance on the test set. Additionally, three integrated learning methods were chosen: Voting, Stacking, and Bagging. The feature selection threshold was adjusted to 0.040, resulting in the retention of five features: time (0.599), mild fatigue occurrences (0.203), number of gaze transfers between AOI and non-AOI (0.049), EAR average (0.047), and moderate fatigue occurrences (0.046). The classifier was then tested with these five features, and it was found that the RF algorithm achieved an accuracy of approximately 97%, while the other algorithms achieved a classification accuracy of roughly 98%. Due to the temporal and incremental nature of the classification task in this study, combined with the analysis results of automatic feature engineering in Featuretools, we calculated the cumulative sum of features with importance greater than 0.01 and incorporated them as a new feature set for the classifiers.

So far, feature set 2 contains the following features:

1. time

2. mild fatigue occurrences

**Table 4. The accuracy of all employed learning algorithms.**

| Algorithm | Cross-Validation Accuracy | Test Accuracy |
| --- | --- | --- |
| SVM | 98.3% | 95.8% |
| DT | 97.2% | 94.4% |
| RF | 97.9% | 97.2% |
| LR | 97.9% | 95.8% |
| KNN | 97.2% | 98.6% |
| NN | 97.9% | 95.8% |
| Voting | 98.6% | 97.2% |
| Stacking | 98.6% | 97.2% |
| Bagging | 97.2% | 98.6% |
| Boosting | 97.2% | 93.1% |

3. number of gaze transfers between AOI and non-AOI

4. EAR average

5. moderate fatigue occurrences

Feature set 3 contains the following features:

1. Cumulative sum of time

2. Cumulative sum of mild fatigue occurrences

3. Cumulative sum of number of gaze transfers between AOI and non-AOI

4. Cumulative sum of EAR average

5. Cumulative sum of moderate fatigue occurrences

6. Cumulative sum of percentage of time spent gazing in AOI0

7. Cumulative sum of M average

8. Cumulative sum of PERCLOS average

As a result, all five machine learning classification algorithms achieved a consistent accuracy of 98.6%. **Table 5** displays the performance comparison among these two new types of feature sets.

Previous studies [67,68] used various combinations of physiological features to assess workload categorization, including electrocardiogram and galvanic skin fusion (acc = 0.74), fusion of multiple physiological signals (heart rate, heart rate variability, electromyography, galvanic skin activity, and respiration) (acc = 0.78), and fusion of physiological signals and task performance (acc = 0.96). The feature combinations used in this study showed significantly improved classification accuracy compared to previous iterations. In addition, five classifiers were evaluated using the complete set of features without any feature selection. The overall classification accuracy achieved with the full set of features was approximately 75%. This comparison demonstrates the efficacy of the feature selection process in this study, as the accuracy is markedly higher when using the selected features. The results indicate that the combination of features identified through the selection process can more accurately describe the workload, thereby enhancing the overall performance of the classifier. This serves to reinforce the value of feature selection methods in improving the accuracy and relevance of workload assessment. However, in this study, the classification algorithm not only helps to evaluate the effectiveness of different types of selected features for classification, but also plays a crucial role in selecting feature sets that accurately characterize the real-time cumulative workload of ramp controllers. In this phase, we analyzed three different combinations of features: those with weights greater

**Table 5. Classification accuracy of feature set 2 and feature set 3 in each algorithm.**

| Algorithm | Cross-Validation Accuracy | | Test Accuracy | |
|---|---|---|---|---|
| | Feature set 2 (weight threshold = 0.04) | Feature set 3 (cumulative sum) | Feature set 2 (weight threshold = 0.04) | Feature set 3 (cumulative sum) |
| RF | 98.2% | 97.9% | 97.2% | 98.6% |
| KNN | 97.2% | 97.2% | 98.6% | 98.6% |
| Voting | 98.6% | 97.9% | 98.6% | 98.6% |
| Stacking | 98.6% | 98.2% | 98.6% | 98.6% |
| Bagging | 97.2% | 97.2% | 98.6% | 98.6% |

than 0.01, those with weights greater than 0.04, and the cumulative sum of features with weights greater than 0.01. The results indicate that the third type of feature set has the highest level of characterization performance.

## 3.4 Numerical workload results

Given the accuracy results of the aforementioned classifiers, we used the third feature set, weighted summation of the features with feature importance for each sample, where the cumulative sum feature uses the same weight as the original feature to convert each unit of time into a numerical value for workload. Cluster 1 consisted of 392 samples with lower workloads. The workload values in this cluster ranged from 0.77 to 1115.45. The median workload value of 199.50 occurred at the 25th minute, while the maximum value occurred at the 60th minute. Cluster 0 consisted of 328 samples with higher workload values ranging from 462.90 to 2523.88. The median workload in this cluster was 1515.83 and occurred at the 79th minute. The minimum workload value occurred in the 38th minute. The analysis of the entire sample showed that the 90th minute had an average value of 2514 and a maximum value of 2524. In contrast, the 1st minute had an average value of 0.81 and a minimum value of 0.77. **Fig 10** shows the numerical and temporal distributions of the workloads, as well as the changes in the workload values over time.

The values of Cluster 1, which is sorted in the bottom 8%, and Cluster 0, which is sorted in the top 8%, are taken as 755.78 and 756.58, respectively, so 756 is chosen as the threshold for lower and higher workloads, which occurs in the 49th minute in this experiment. **Fig 11** is a simple graphical illustration of the workload threshold determination method where the workload values for each 10% place in the two workload levels are labeled.

In this study, ramp controller workload is converted into real-time numerical values and compared to previous studies that only assessed personnel workload grades [69–72], this method provides continuous monitoring of ramp controller workload and shows the accumulation of workload, allowing airport managers to identify potential human factor risks, detect trends in a timely manner before personnel reach workload thresholds or overload conditions, rationalize the use of personnel and resources to safeguard critical tasks, and improve the science of decision making.

It is acknowledged that investigating the relationship between workload and task performance represents a valuable contribution to the field. The principal objective of our study was to identify and quantify workload through the utilization of a detection system. Although the immediate impact of high workload on task performance may not be readily apparent, our findings indicate a significant decline in performance over time. Specifically, we employed the ratio of instruction omissions or errors as the performance metric. While the short-term effects of high workload on task performance were not immediately evident, a notable deterioration was observed during prolonged high workload conditions, with the error rate increasing from 5% to over 15%. This further supports the effectiveness of our workload detection system, which provides critical early warnings in high workload situations. These early warnings allow for timely adjustments in task scheduling and workflow optimization, ultimately helping to maintain overall task performance and prevent long-term degradation.

## 3.5 Practical implication

The practical implications of this study are twofold, highlighting its significance in the domain of aviation safety and workload management for ramp controllers. Firstly, by integrating fatigue, eye movement, and respiratory characteristics into a workload assessment model and calibrating the optimal set of feature combinations, this research offers a novel approach to

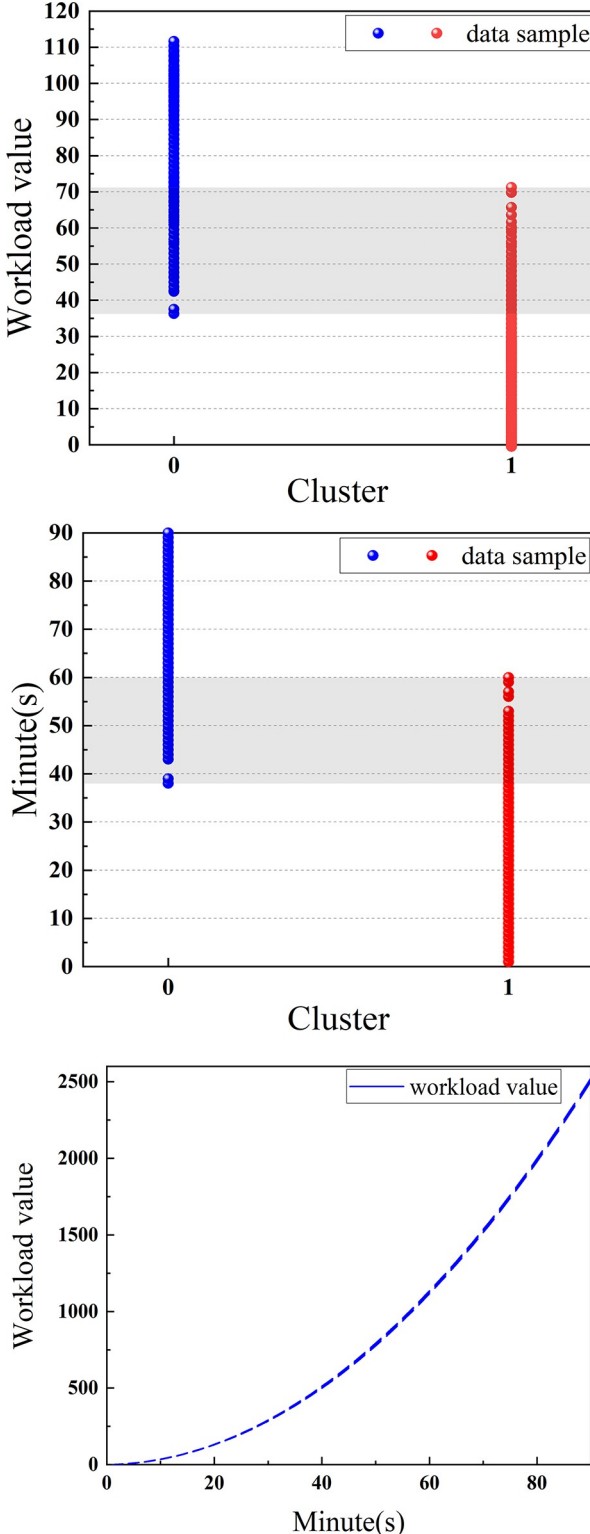

**Fig 10. Workload numerical and temporal distributions, and the changes in the workload values over time.** The gray portion indicates the overlapping intervals of the samples in the two workload levels in both numerical and temporal dimensions.

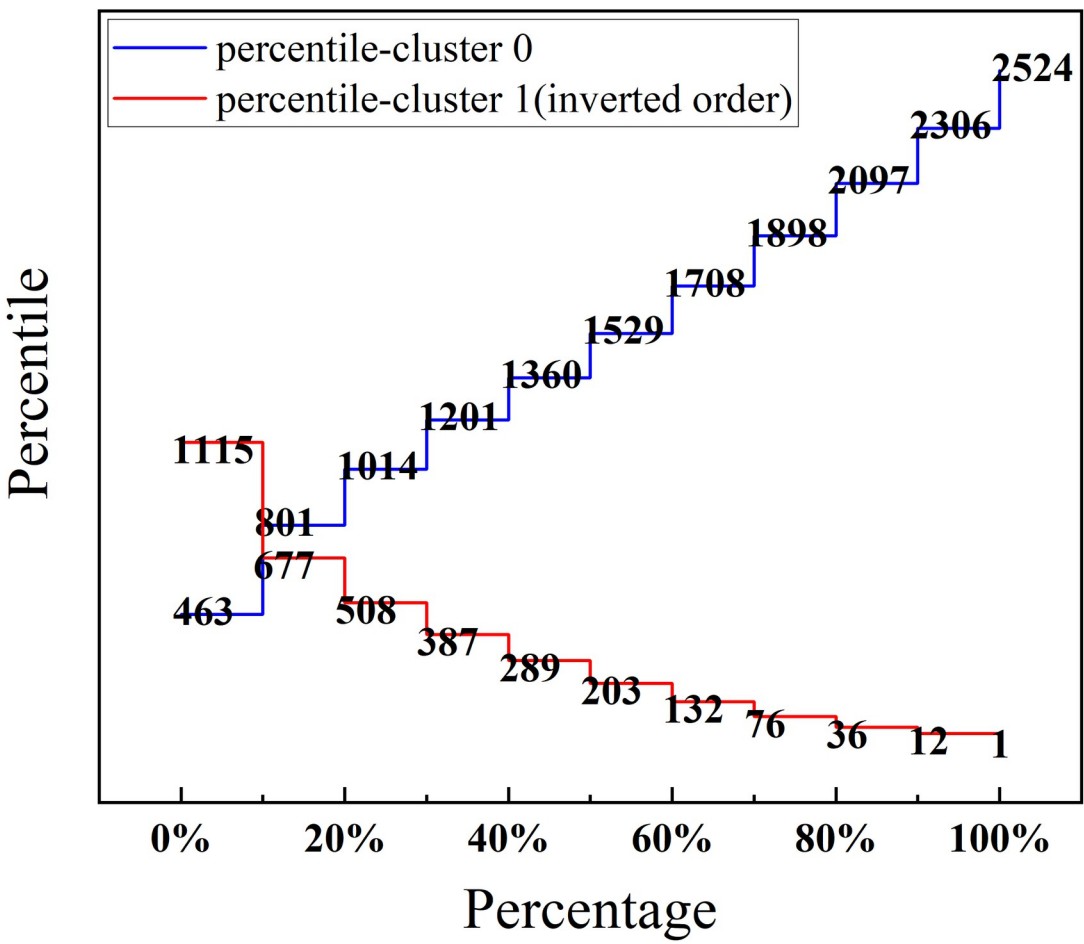

**Fig 11. Workload threshold determination method.** The class thresholds for the workloads are between 677–801 and are sorted in the 5%-10% range, and the search continues within that range until a unique value is found.

enhancing the real-time numerical evaluation of workload for airport ramp controllers. Ramp controllers must manage their workload effectively while handling complex information during control operations to ensure aviation safety. Accurately detecting their instantaneous workload is crucial for maintaining control effectiveness and preventing hazardous incidents. This new method facilitates a more efficient assessment of cumulative workload under high-stress conditions, enabling the timely identification of excessive workloads. Consequently, measures can be implemented to alleviate the burden, thereby minimizing the likelihood of operational errors due to fatigue or overload, significantly enhancing aviation safety.

Second, the application of this research transcends the immediate context of ramp control operations, offering potential benefits to a broader range of high-stress professions where workload management is critical for performance and safety. The methodology developed through this study—combining eye-tracking data, respiratory patterns, and fatigue indicators into a comprehensive workload assessment model—provides a blueprint for similar applications in other fields such as air traffic control, emergency response teams, and healthcare. By enabling a more nuanced understanding of how various physiological and psychological indicators correlate with workload levels, this approach facilitates the development of targeted interventions to mitigate risks associated with high workload scenarios.

In addition, the use of advanced machine learning techniques to calibrate the optimal set of feature combinations for workload assessment marks a significant advancement in predictive analytics within the field of occupational health and safety. The high classification accuracy achieved in this study demonstrates the feasibility of deploying such models in real-world settings to dynamically assess workload in real time. This capability paves the way for the development of intelligent systems that can provide immediate feedback to both individuals and management, allowing for the swift implementation of corrective measures to prevent overload and its associated risks.

Overall, this study not only contributes to the enhancement of aviation safety through improved workload management for ramp controllers but also sets a precedent for the application of sophisticated data analytics techniques in occupational health and safety management across various industries. By harnessing the power of data-driven insights to understand and mitigate workload-related risks, this research underscores the potential of technology to transform workplace safety and efficiency in the 21st century.

### 3.6 Future research

Future research can integrate this method into the ramp controller shift rotation program. In terms of practical application, the load thresholds can be evaluated separately based on the individual differences of the personnel within each shift of ramp controllers at a given airport, combining their work habits, personality traits, etc., and the load thresholds can be used as the basis for shift rotation, so that the level of both safety and efficiency of ramp control can be improved.

Furthermore, future studies could examine the phenomenon of inattentional deafness, whereby individuals are unable to hear alarm messages that appear when severe fatigue is detected, as highlighted by Durantin et.al [73]and Callan et.al [74]. An investigation into the impact of fatigue on operators' ability to maintain attention, coupled with an evaluation of potential remedies, such as alternative alerting methods, could markedly enhance the efficacy of fatigue detection systems.

Moreover, a subsequent study could compare the efficacy of different types of workload displays in assisting operators to remain focused on the task at hand. For example, Azey et.al [75] demonstrated the potential of cognitive workload assessments using optical brain imaging sensors, which could be applied to compare the effectiveness of various workload displays in real-time operational environments.

Finally, a more thorough investigation into the relationship between EAR, PERCLOS, and workload is essential. While these metrics are typically linked to drowsiness and underload scenarios, validating their efficacy as reliable indicators of overload within this context is paramount. Exploring alternative facial tracking techniques or integrating additional biometric data could further enhance the accuracy of workload assessment in future implementations.

### 4. Conclusions

This paper presents a numerical representation of the real-time workload of ramp controllers by combining various characteristics (eye tracker, respiratory, and fatigue patterns). In the experiment, a facial and eye recognition-based system was developed and used to monitor the real-time fatigue status of the ramp controllers. Eye movement and respiration data were also collected. The collected three kinds of feature data were clustered using the K-means, and further feature weighting and classifier training were completed using random forest and other supervised learning methods. The identified processes confidently selected the most appropriate feature combination to characterize the ramp controller's workload, improving the

accuracy of the real-time workload assessment model and providing valuable guidance for metric selection in future research. Following feature weight values, feature combination construction methods and workload level labels, the study developed a numerical expressive method for workload values and a calculation method for workload level thresholds. The research provides valuable insights for developing safety intelligence in the civil aviation industry. This approach can be implemented and extended in real-world environments to improve its reliability and usefulness.

## Supporting information

**S1 Dataset.**
(PDF)

## Author Contributions

**Conceptualization:** Quan Shao, Kaiyue Jiang.

**Data curation:** Quan Shao, Kaiyue Jiang.

**Formal analysis:** Kaiyue Jiang, Ruoheng Li.

**Funding acquisition:** Quan Shao.

**Investigation:** Kaiyue Jiang.

**Methodology:** Kaiyue Jiang.

**Project administration:** Quan Shao.

**Resources:** Quan Shao, Kaiyue Jiang.

**Software:** Kaiyue Jiang.

**Supervision:** Quan Shao, Kaiyue Jiang.

**Validation:** Kaiyue Jiang.

**Visualization:** Kaiyue Jiang.

**Writing – original draft:** Kaiyue Jiang.

**Writing – review & editing:** Kaiyue Jiang, Ruoheng Li.

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
