## [Decision Letter · Decision Letter 0]

5 Aug 2024

PONE-D-24-23960A numerical evaluation of real-time workloads for ramp controller through optimization of multi-type feature combinations derived from eye tracker, respiratory, and fatigue patternsPLOS ONE

Dear Dr. Shao,

Thank you for submitting your manuscript to PLOS ONE. After careful consideration, we feel that it has merit but does not fully meet PLOS ONE’s publication criteria as it currently stands. Therefore, we invite you to submit a revised version of the manuscript that addresses the points raised during the review process.

We look forward to receiving your revised manuscript.

Kind regards,

Hyuk Oh

Academic Editor

PLOS ONE

Journal Requirements:

2. Thank you for stating the following financial disclosure: "This work was supported by National Natural Science Foundation of China-Civil Aviation Administration of China Civil Aviation Joint Research Fund Project (U2233208)."

3. Thank you for stating the following in the Acknowledgments Section of your manuscript: "This work was supported by National Natural Science Foundation of China-Civil Aviation Administration of China Civil Aviation Joint Research Fund Project (U2233208)."

Please remove any funding-related text from the manuscript and let us know how you would like to update your Funding Statement. Currently, your Funding Statement reads as follows: "This work was supported by National Natural Science Foundation of China-Civil Aviation Administration of China Civil Aviation Joint Research Fund Project (U2233208)."

4. PLOS requires an ORCID iD for the corresponding author in Editorial Manager on papers submitted after December 6th, 2016. Please ensure that you have an ORCID iD and that it is validated in Editorial Manager. To do this, go to ‘Update my Information’ (in the upper left-hand corner of the main menu), and click on the Fetch/Validate link next to the ORCID field. This will take you to the ORCID site and allow you to create a new iD or authenticate a pre-existing iD in Editorial Manager. Please see the following video for instructions on linking an ORCID iD to your Editorial Manager account: https://www.youtube.com/watch?v=_xcclfuvtxQ.

5. Please amend your authorship list in your manuscript file to include author Dr. Ruoheng Li.

7. We note that Figures 2 and 4 includes an image of a participant in the study. 

Reviewers' comments:

Reviewer's Responses to Questions

**Comments to the Author**

1. Is the manuscript technically sound, and do the data support the conclusions?

Reviewer #1: Partly

Reviewer #2: Yes

2. Has the statistical analysis been performed appropriately and rigorously? 

Reviewer #1: Yes

Reviewer #2: Yes

3. Have the authors made all data underlying the findings in their manuscript fully available?

Reviewer #1: No

Reviewer #2: Yes

4. Is the manuscript presented in an intelligible fashion and written in standard English?

Reviewer #1: Yes

Reviewer #2: Yes

5. Review Comments to the Author

Reviewer #1: The authors present a study on adopting feature selection techniques with classifiers to evaluate ramp controllers' real-time workloads. The paper has provided a lot of fundamental concepts around the background, motivation, and adopted techniques.

Please find below suggestions for improving the work:

1. In the introduction section, there are many duplicative sentences, and they can be merged into one. For example, lines 61 to 65, 72 to 76, 89 to 94, etc. Please update the whole manuscript to avoid duplicated contents.

2. There are sentences that need to include in-text citations, such as lines 70-72, 432, etc. Please also revise and check the whole manuscript.

3. The labeling of fatigue level is not clear enough. What do you mean by “The consecutive frames 5, 15, and 50 are used to classify the fatigue level as mild, moderate, or severe”? This needs more clarifications.

4. The experimental setup (i.e., simulation task setup, experimental trials, and fatigue level count) needs more clarification. How many trials did the authors perform? For how long each? How did you make sure the participant experience fatigue? A flowchart or an experimental setup illustration is required.

5. Fatigue can also be seen as a state that needs to be detected/assessed, yet the authors choose to make it as an attribute/feature for workload evaluation. Is this appropriate?

6. The authors claim “In addition, a separate test set comprising 20% of the dataset was randomly selected for model comparison. The remaining data was used for 5-fold cross-validation.” What is the motivation of doing this? Is this even necessary? Why do not simply go for cross-validation? BTW, how did you select the results of the 20% testing set? Through performing repeated experiments and select the most accurate one?

7. The distribution of the three-level fatigue is not presented. Among your 720 instances, how many instances are with mild, moderate, severe classes? The same for the workload, the distribution is not involved.

8. Fig. 5 is very confusing. The feature set selection module is not involved in any further processing, I’m assuming this figure is not correct. And it is not a correct flowchart demonstration, please use the correct shape for process, data, and other interactions.

9. Feature set 2 and feature set 3 should be independently presented with tables or lists. Currently, it's quite confusing about the differences between them.

10. Please include classification results with all features, and have them compared to the results with selected features. Then please provide discussions accordingly.

11. Since there is no literature summary table, it is better if the authors could provide a performance comparison table to list the results of relevant existing studies and current work.

Reviewer #2: • What are the main claims of the paper and how significant are they for the discipline?

The authors have designed an experiment to collect realistic ramp controller data from experienced operators using real data from Shanghai Hongqiao Airport. Three distinct types of workload correlates were collected throughout the 90 minute experiment, focusing on eye tracking, respiration, and facial features/expressions. This data was analyzed using a two-step process with unsupervised learning followed by a comparison of supervised learning techniques to create the first steps of a real-time overload detection system. The multimodal fusion used here will be helpful for the community as an example for more unique combinations in the future.

• Are the claims properly placed in the context of the previous literature? Have the authors treated the literature fairly?

There is a slightly high reliance on a small number of papers in the literature to back up the authors’ claims, but I have suggested several key papers from the literature to add to the publication to strengthen the background.

• Do the data and analyses fully support the claims? If not, what other evidence is required?

The major claim of the paper to develop a method for classifying cumulative workload in ramp controllers appears to have been met. However, performance measures were not analyzed and presented in the current publication, so it is unclear whether the detected workload states are actually resulting in a meaningful decrease in the ability to complete the task successfully. If performance did not suffer at moments of high workload, that should be mentioned. If performance did change, such as a decrease over time, this must be incorporated into the findings.

• PLOS ONE encourages authors to publish detailed protocols and algorithms as supporting information online. Do any particular methods used in the manuscript warrant such treatment? If a protocol is already provided, for example for a randomized controlled trial, are there any important deviations from it? If so, have the authors explained adequately why the deviations occurred?

As presented, each technique individually is already known in the literature, and more unique aspects such as the M-value calculation are given enough information to replicate in the present form.

• If the paper is considered unsuitable for publication in its present form, does the study itself show sufficient potential that the authors should be encouraged to resubmit a revised version?

With the suggested revisions and answers, I believe this paper will be suitable for publication.

• Are original data deposited in appropriate repositories and accession/version numbers provided for genes, proteins, mutants, diseases, etc.?

The supplementary information contains the processed data, allowing for other researchers to run their own analyses.

• Does the study conform to any relevant guidelines such as CONSORT, MIAME, QUORUM, STROBE, and the Fort Lauderdale agreement?

These are followed or otherwise do not apply.

• Are details of the methodology sufficient to allow the experiments to be reproduced?

Yes, I believe this experiment is reproducible based on the supplied methodology.

• Is any software created by the authors freely available?

I have asked about the facial recognition software in my comments below. If it is created by the authors, I do not currently see it, but it may be publicly available.

• Is the manuscript well organized and written clearly enough to be accessible to non-specialists?

I have asked for several clarifying edits to be made in my comments below to improve the flow and understandability. If these changes are made, it will be clear.

• Is it your opinion that this manuscript contains an NIH-defined experiment of Dual Use concern?

No, this research is not Dual Use.

Section 1.1 could use more references to cognitive workload and neuroergonomics, as well as ECG and EEG, eye tracking.

Ayaz, H., et al. (2012). "Optical brain monitoring for operator training and mental workload assessment." Neuroimage 59(1): 36-47.

Mehta, R. K. and R. Parasuraman (2013). "Neuroergonomics: A Review of Applications to Physical and Cognitive Work." Frontiers in Human Neuroscience 7.

Ahlstrom, U. and F. J. Friedman-Berg (2006). "Using eye movement activity as a correlate of cognitive workload." International Journal of Industrial Ergonomics 36(7): 623-636.

Ayaz, H., et al. (2011). Estimation of Cognitive Workload during Simulated Air Traffic Control Using Optical Brain Imaging Sensors. Foundations of Augmented Cognition. Directing the Future of Adaptive Systems: 6th International Conference, FAC 2011, Held as Part of HCI International 2011, Orlando, FL, USA, July 9-14, 2011. Proceedings. D. D. Schmorrow and C. M. Fidopiastis. Berlin, Heidelberg, Springer Berlin Heidelberg: 549-558.

Mark, J. A., et al. (2024). "Mental workload assessment by monitoring brain, heart, and eye with six biomedical modalities during six cognitive tasks." Frontiers in Neuroergonomics 5.

Section 1.2 workload definition can use more than Hart 1988 (NASA-TLX).

Wickens, C. D. (2002). "Multiple resources and performance prediction." Theoretical Issues in Ergonomics Science 3(2): 159-177.

Cain, B. (2007). "A review of the mental workload literature." DTIC Document.

Hancock, P. and M. H. Chignell (1986). "Toward a Theory of Mental Work Load: Stress and Adaptability in Human-Machine Systems." Proc. IEEE SMC 1986: 378-383.

Section 1.3 needs citations for the first claim that physiological features are used to assess workload.

Scerbo, M., et al. (2001). "The Efficacy of Psychophysiological Measures for Implementing Adaptive Technology."

Was a specific software used for the facial fatigue recognition, or did the authors create it? Please clarify this in the text.

Section 2.2.2, are the monitor sizes and distances standard and comparable to the real world equivalent for a ramp controller? This is important for the ecological validity of the results.

During the 90 minute task, how long was the average trial and was there variation, what counted as a single trial, and also were there any breaks?

Sections 2.2.2 and 2.3.1 discuss the frames used in the fatigue calculations, but no mention of framerate is given. Please clarify the recording frequency and also add the length of time (preferably in ms) for how long 5, 15, and 50 frames are that are used to calculate fatigue level.

It may benefit the paper to discuss inattentional deafness to the alarm messages that appear when severe fatigue is detected, and how this may be remedied when fatigue prevents the operators from fixing their attention.

Durantin, G., et al. (2017). "Neural signature of inattentional deafness." Hum Brain Mapp 38(11): 5440-5455.

Callan, D. E., et al. (2023). "The role of brain-localized gamma and alpha oscillations in inattentional deafness: implications for understanding human attention." Frontiers in Human Neuroscience 17: 1168108.

Is there a benefit to displaying a live camera feed of the operator to them in real time? In actual use, I would expect just the warnings or fatigue levels to be displayed. Please discuss a bit about the reasoning for this choice.

Regarding these last two points, a followup study may compare different types of displays and compare what workload information best help operators stay on task, such as here:

Ayaz, H., et al. (2010). Cognitive Workload Assessment of Air Traffic Controllers Using Optical Brain Imaging Sensors. Advances in Understanding Human Performance, CRC Press: 21-31.

You mention that the Eyeso Glasses interfere with the fatigue measurements of the eye, have you considered a mounted eye tracker rather than a facial worn one? Why did you not use one?

Section 2.4 what methods were used to clean the data, and was there a method for determining which low variation measures to eliminate? Was there an arbitrary cutoff, did you use PCA, etc. I am not referring to the PCA conducted after feature selection.

Section 3.1, you claim that EAR and PERCLOS, both measures of fatigue, directly correlate with overload condition. However, drowsiness generally correlates with the opposite, underload. How do you validate that this was the best facial tracking measure for workload?

Sections 3.2, it was not immediately clear to me the process of selecting features from each of the three categories, and then running PCA to create two principal components which were then used in k-clustering for the unsupervised learning step. This could be made more clear with some added explanation in Section 3.3.

It appears that the main takeaway and application is to find the moment in time where operators cross over from low to high workload and need a break in order to recover before continuing. However, this is only important if the actual task performance decreases as a result of the workload measurements. Did you take any behavioral performance measures to demonstrate this point?

6. PLOS authors have the option to publish the peer review history of their article (what does this mean?). If published, this will include your full peer review and any attached files.

Reviewer #1: No

Reviewer #2: **Yes**

---

## [Author Response · Author response to Decision Letter 0]

1 Sep 2024

I sincerely appreciate your valuable feedback on my study. I have carefully addressed all your comments, and you can kindly find my detailed responses in the "Respond to Reviewers" document. Thank you for your thoughtful insights.

---

## [Decision Letter · Decision Letter 1]

8 Oct 2024

PONE-D-24-23960R1A numerical evaluation of real-time workloads for ramp controller through optimization of multi-type feature combinations derived from eye tracker, respiratory, and fatigue patternsPLOS ONE

Dear Dr. Shao,

Thank you for submitting your manuscript to PLOS ONE. After careful consideration, we feel that it has merit but does not fully meet PLOS ONE’s publication criteria as it currently stands. Therefore, we invite you to submit a revised version of the manuscript that addresses the points raised during the review process.

We look forward to receiving your revised manuscript.

Kind regards,

Hyuk Oh

Academic Editor

PLOS ONE

**Journal Requirements:**

**Additional Editor Comments:**

Follow the formatting guidelines and proofread for acceptance as requested by a reviewer.

Reviewers' comments:

Reviewer's Responses to Questions

**Comments to the Author**

1. If the authors have adequately addressed your comments raised in a previous round of review and you feel that this manuscript is now acceptable for publication, you may indicate that here to bypass the “Comments to the Author” section, enter your conflict of interest statement in the “Confidential to Editor” section, and submit your "Accept" recommendation.

Reviewer #1: (No Response)

Reviewer #2: All comments have been addressed

2. Is the manuscript technically sound, and do the data support the conclusions?

Reviewer #1: Yes

Reviewer #2: Yes

3. Has the statistical analysis been performed appropriately and rigorously? 

Reviewer #1: Yes

Reviewer #2: Yes

4. Have the authors made all data underlying the findings in their manuscript fully available?

Reviewer #1: Yes

Reviewer #2: Yes

5. Is the manuscript presented in an intelligible fashion and written in standard English?

Reviewer #1: Yes

Reviewer #2: Yes

6. Review Comments to the Author

**Reviewer #1:** Thanks to the authors for putting effort in revising this manuscript, I find the revised version more solid and clear. And I'm happy with most of their responses.

But I do have a few minor suggestions for the authors to improve before the paper gets published.

Please see my comments here:

1. Please correct the in-text citation styles. For example, in line 68, it should be "He et al. [6]" rather than "[6]". Please check the entire manuscript.

2. When use in-text citations, there should be a space after the sentence. For example, in line 77, it should be "... the cardiac cycle [8-10]" and "... correlating workload states [11,12]" in line 80. Please check the entire manuscript.

3. For existing literature comparison (Table 1), it is expected to list the methodology peformance.

4. The authors should carefuly go through proofreading before the paper gets accepted. There are many typo mistakes that need to be revised.

Hope this will help the authors to form into a more acceptable version.

**Reviewer #2: **(No Response)

7. PLOS authors have the option to publish the peer review history of their article (what does this mean?). If published, this will include your full peer review and any attached files.

Reviewer #1: No

Reviewer #2: No

---

## [Author Response · Author response to Decision Letter 1]

19 Oct 2024

Thank you for your feedback. For detailed responses to your comments, please refer to the “Response to Reviewers-R2” document. We appreciate your time and effort in reviewing our manuscript.

---

## [Editor Report · Decision Letter 2]

28 Oct 2024

A numerical evaluation of real-time workloads for ramp controller through optimization of multi-type feature combinations derived from eye tracker, respiratory, and fatigue patterns

PONE-D-24-23960R2

Dear Dr. Shao,

We’re pleased to inform you that your manuscript has been judged scientifically suitable for publication and will be formally accepted for publication once it meets all outstanding technical requirements.

Kind regards,

Hyuk Oh

Academic Editor

PLOS ONE
---

## [Editor Report · Acceptance letter]

30 Oct 2024

PONE-D-24-23960R2 

PLOS ONE

Dear Dr. Shao, 

I'm pleased to inform you that your manuscript has been deemed suitable for publication in PLOS ONE. Congratulations! Your manuscript is now being handed over to our production team.

Kind regards, 

on behalf of

Dr. Hyuk Oh 

Academic Editor

PLOS ONE